# Fibroblast Activation Protein Expressing Mesenchymal Cells Promote Glioblastoma Angiogenesis

**DOI:** 10.3390/cancers13133304

**Published:** 2021-07-01

**Authors:** Eva Balaziova, Petr Vymola, Petr Hrabal, Rosana Mateu, Michal Zubal, Robert Tomas, David Netuka, Filip Kramar, Zuzana Zemanova, Karla Svobodova, Marek Brabec, Aleksi Sedo, Petr Busek

**Affiliations:** 1Laboratory of Cancer Cell Biology, Institute of Biochemistry and Experimental Oncology, First Faculty of Medicine, Charles University, 128 00 Prague, Czech Republic; eva.balaziova@lf1.cuni.cz (E.B.); petr.vymola@lf1.cuni.cz (P.V.); rosana.mateu@lf1.cuni.cz (R.M.); michal.zubal@lf1.cuni.cz (M.Z.); 2Department of Pathology, Military University Hospital, 169 02 Prague, Czech Republic; petr.hrabal@uvn.cz; 3Departments of Neurosurgery, Na Homolce Hospital, 150 00 Prague, Czech Republic; robert.tomas@homolka.cz; 4Department of Neurosurgery and Neurooncology, First Faculty of Medicine, Charles University and Military University Hospital, 168 02 Prague, Czech Republic; david.netuka@uvn.cz (D.N.); filip.kramar@uvn.cz (F.K.); 5Center of Oncocytogenomics, Institute of Clinical Biochemistry and Laboratory Diagnostics, General University Hospital and First Faculty of Medicine, Charles University, 128 00 Prague, Czech Republic; zuzana.zemanova@vfn.cz (Z.Z.); karla.svobodova@vfn.cz (K.S.); 6Institute of Computer Science, The Czech Academy of Sciences, 128 00 Prague, Czech Republic; mbrabec@cs.cas.cz

**Keywords:** glioblastoma, angiogenesis, microenvironment, fibroblast activation protein, seprase, angiopoietin, vessel destabilisation

## Abstract

**Simple Summary:**

The perivascular niche in glioblastoma is crucial for maintaining a tumour- permissive microenvironment. In various extracranial cancers, mesenchymal cells that express fibroblast activation protein (FAP) are an important stromal component and a potential therapeutic target. In this study, we examine their functions in the glioblastoma microenvironment where their role is so far largely unexplored. Glioblastoma-associated FAP^+^ mesenchymal cells are localised around activated endothelial cells and their presence positively correlates with vascular density. They represent a subpopulation of stromal, non-tumorigenic cells which mostly lack the chromosomal aberrations characteristic of glioma cells. By soluble factors they induce angiogenic sprouting, chemotaxis of endothelial cells, contribute to destabilisation of blood vessels, and increase the migration and growth of glioma cells. Taken together, we identified a subpopulation of FAP^+^ mesenchymal cells in the perivascular niche in glioblastoma that may contribute to tumour progression by promoting angiogenesis and supporting dissemination of transformed cells into the surrounding tissue.

**Abstract:**

Fibroblast activation protein (FAP) is a membrane-bound protease that is upregulated in a wide range of tumours and viewed as a marker of tumour-promoting stroma. Previously, we demonstrated increased FAP expression in glioblastomas and described its localisation in cancer and stromal cells. In this study, we show that FAP^+^ stromal cells are mostly localised in the vicinity of activated CD105^+^ endothelial cells and their quantity positively correlates with glioblastoma vascularisation. FAP^+^ mesenchymal cells derived from human glioblastomas are non-tumorigenic and mostly lack the cytogenetic aberrations characteristic of glioblastomas. Conditioned media from these cells induce angiogenic sprouting and chemotaxis of endothelial cells and promote migration and growth of glioma cells. In a chorioallantoic membrane assay, co-application of FAP^+^ mesenchymal cells with glioma cells was associated with enhanced abnormal angiogenesis, as evidenced by an increased number of erythrocytes in vessel-like structures and higher occurrence of haemorrhages. FAP^+^ mesenchymal cells express proangiogenic factors, but in comparison to normal pericytes exhibit decreased levels of antiangiogenic molecules and an increased Angiopoietin 2/1 ratio. Our results show that FAP^+^ mesenchymal cells promote angiogenesis and glioma cell migration and growth by paracrine communication and in this manner, they may thus contribute to glioblastoma progression.

## 1. Introduction

Glioblastomas (GBMs) are aggressive malignancies with dismal prognosis. These highly heterogeneous tumours are characterised by rapid and invasive growth leading to diffuse infiltration of the surrounding tissue and rampant neovascularisation [1] with microvascular proliferations and pseudopalisading necroses stemming from occlusion of blood vessels [2]. Aberrant angiogenesis in the GBM microenvironment is driven by several proangiogenic mediators, such as vascular endothelial growth factor (VEGF). Additionally, an imbalance between regulators of angiogenesis–exemplified by Angiopoietin-1 (Ang-1) and Angiopoietin-2 (Ang-2)–contributes to the destabilisation of the newly formed blood vessels [3,4].

GBM progression is propelled by growth-promoting alterations in the p53, RB, and PI3K pathways in cancer cells [5] and by extensive crosstalk between the cancer cells and non-malignant stromal cells. It has been shown that immune and endothelial cells, neurons, and astrocytes all play an important role in stimulating GBM growth and angiogenesis [6]. There is moreover a growing body of evidence that different types of mesenchymal cells may contribute to GBM progression. We and others [7,8,9] have previously demonstrated that the expression of fibroblast activation protein (FAP), a characteristic marker of tumour-associated mesenchymal cells [10], is increased in GBM. The highest expression of this membrane-bound serine protease was observed in the mesenchymal subtype of GBM, which is characterised by more frequent necroses and overexpression of angiogenesis markers [11,12]. Although in some GBMs FAP is expressed in transformed cells, its expression is mostly confined to stromal cells, in particular mesenchymal cells located around blood vessels and a subpopulation of endothelial cells [8,9,13]. FAP-expressing stromal cells in the perivascular niche are surrounded by fibrillar extracellular matrix proteins, such as fibronectin, and express mesenchymal markers [8] consistent with the phenotype of pericytes and/or cancer-associated fibroblasts. GBM perivascular niche is a compartment in which angiogenesis is initiated and self-renewing brain tumour cells are localised and maintained in a stem cell-like state [14]. Moreover, the modified extracellular matrix and a cocktail of motility-stimulating mediators present in the perivascular niche provide support for the characteristic perivascular spreading of malignant cells [15,16,17].

In several tumour types, FAP^+^ stromal cells contribute to immunosuppression [18], cancer cell growth [14,19], and it has been proposed that they also contribute to angiogenesis [20,21,22]. Still, their role in the pathogenesis of human GBM remains largely unknown. The aim of this study was to assess the role of FAP^+^ stromal cells in the GBM microenvironment.

## 2. Materials and Methods

### 2.1. Patients, Survival Analysis 

Patients with newly diagnosed GBM (*n* = 94) undergoing neurosurgical resection at the Department of Neurosurgery, Hospital Na Homolce in Prague and Department of Neurosurgery and Neurooncology, the First Faculty of Medicine, Charles university and Military University Hospital Prague. The study was approved by the Institutional Ethics Committee (study approval numbers 108-39/4-2014-UVN and 7/8/2014-25) and conducted in accordance with the Helsinki Declaration. Full informed consent was obtained from all donors prior to neurosurgical resection. GBM diagnosis was established according to the current WHO classification [1], patients received no preoperative oncological treatment. For survival analysis, four patients were excluded due to early postoperative morbidity/mortality or the presence of other cancer. IDH mutation was detected only in 4.4% (4/90) of patients and was therefore not used for further stratification of patients. Characteristics of the patient cohort used for survival analysis are listed in Table 1. Patients underwent standard treatment including maximum safe resection and adjuvant therapy as recommended by the treating physicians. Adjuvant therapy included standardly used Stupp protocol [23] in 40% (36/90) of patients, chemoradiotherapy without adjuvant chemotherapy in 20% (18/90), radiotherapy alone in 11% (10/90), and no therapy in 8% (7/90) of patients. Progression-free survival (PFS) was defined as the interval from surgery to the first evidence of disease progression based on MRI (in 68 patients) or a neurological examination (in 3 patients), or the date of the last follow-up or death depending on which was earlier [24]. Data were analysed by multiple logistic regression and multiple Cox regression models [25].

### 2.2. Cell Lines and Cell Culture 

Unless specified otherwise, cell culture was performed under standard conditions at 37 °C in a humidified atmosphere of 5% CO_2_ in air. Glioma cell lines U251 and U87 were obtained from ATCC (LGC Standards, Teddington, UK) and cultured in Dulbecco’s Modified Eagle’s Medium (DMEM, Merck, Darmstadt, Germany) supplemented with 10% foetal calf serum (FCS, Merck). To prepare glioma cells that would constitutively express a fluorescent protein, U251 cells were transfected in a six-well plate with 2 μg of pEGFP-N2 plasmid encoding enhanced green fluorescent protein (Addgene, Teddington, UK) using Lipofectamine 3000 Transfection Reagent (Thermo Fisher Scientific, Waltham, MA, USA) for 4–6 h following the manufacturer’s protocol. To select stable clones, we used Geneticin (G418, Merck) at a concentration of 400 µg/mL. Human umbilical vein endothelial cells (HUVEC, Thermo Fisher Scientific) were cultured according to manufacturer’s recommendations in M200 medium with 2% large vessel endothelial supplement (LVES, Thermo Fisher Scientific). Human brain vascular pericytes (HBVP) were acquired from ScienCell (ScienCell Research Laboratories, Carlsbad, CA, USA) and cultured under conditions recommended by the provider in pericyte medium (PM) supplemented with 2% foetal bovine serum (FBS), pericyte growth supplement (PGS), 100 units/mL penicillin G, and 100 µg/mL streptomycin (P/S, all provided by ScienCell Research Laboratories).

### 2.3. Derivation of Primary Cell Cultures

FAP^+^ mesenchymal cells were derived from fresh human GBM tissue. Tumour tissue was cut into small pieces, digested with TrypLE Select (Thermo Fisher Scientific) at 37 °C for 20 min and placed on fibronectin-coated (3 μg/cm^2^, Merck) plastic and cultured in PM supplemented 2% FBS, PGS, and P/S (ScienCell Research Laboratories) for seven days. Afterwards, the cells migrating from the explants were harvested and used for indirect magnetic-activated cell sorting (MACS) according to manufacturer’s instructions with minor modifications. Briefly, the cells were resuspended at 10 × 10^6^/mL and incubated with 0.1 μg of anti-FAP F11-24 antibody (Santa Cruz Biotechnology, Dallas, TX, USA) per 1 × 10^6^ cells at 4 °C for 8 min. After two washes, the labelled cells were resuspended at 10 × 10^6^/mL, then 0.15 μL of magnetic DynaBeads (Thermo Fisher Scientific) were added per 1 × 10^6^ cells, and the resulting suspension was incubated at 4 °C for 4 min. The FAP^+^ fraction was separated and cultured on fibronectin-coated plastic in PM supplemented with 2% FBS, PGS, and P/S. Alternatively, FAP^+^ fraction was isolated directly from the cell suspension after mechanical and enzymatic digestion.

An analogous procedure was used to derive primary microvascular endothelial cell cultures from fresh human GBM tissue. Mechanically and enzymatically digested tissue fragments were cultured on fibronectin-coated plastic in an initiation medium containing RPMI-1640, 10% FCS, 10% Nu serum (BD Bioscience, Franklin Lakes, NJ, USA), 1 mM HEPES (Merck), 300 UI Heparin (Zentiva, Prague, Czech Republic), 1% P/S (Merck), and 3 µg/mL endothelial cell growth supplement (ECGS, BD Bioscience) [26]. Afterwards, the cells migrating from the explants were harvested and used for indirect magnetic-activated cell sorting (MACS) using an anti-CD105 antibody (M3527, Dako Agilent, Santa Clara, CA, USA, 0.15 µg per 1 × 10^6^ cells at 4 °C for 8 min). The positive fraction was cultured on fibronectin-coated plastic in an initiation medium. After two weeks, 10% Nu Serum was omitted from the culture media and ECGS concentration was increased to 30 µg/mL [26]. Expression of an endothelial cell marker von Willebrand factor and absence of glial (GFAP) and mesenchymal (TE-7) markers was confirmed by immunocytochemistry as described below. Experiments using primary cell cultures (FAP^+^ mesenchymal cells and primary microvascular endothelial cells) were performed at passage two to five to preserve the original characteristics of the cells. 

### 2.4. Preparation of a Conditioned Medium

To obtain serum-free conditioned media, 1 × 10^6^ FAP^+^ mesenchymal cells and HBVP, respectively, were seeded in 100 mm dishes and grown in PM supplemented with 2% FBS, PGS, and P/S for 24 h. Then the medium was removed, cells washed twice with PBS, and 10 mL of serum-free PM were added. Cells were cultured at 37 °C under standard conditions (normoxic conditioned media) or in an airtight humidified hypoxic chamber (STEMCELL Technologies, Vancouver, BC, Canada) containing 1% O_2_, 5% CO_2_, and 94% N_2_ (hypoxic conditioned media). After 72 h, the medium was collected, centrifuged (618× *g* at 4 °C for 10 min), and the supernatant filtered through a 0.22 μm pore filter (Merck) and stored in aliquots at −75 °C until use. To prepare a HUVEC conditioned medium, 0.3 × 10^6^ HUVEC were seeded in 60 mm dishes in M200 medium supplemented with 2% LVES. After 72 h, cells were washed twice with PBS and medium was replaced with M200 medium supplemented with 1% FCS. Conditioned media were collected after 48 h and processed as described above. Corresponding media not exposed to cells were used as controls. 

### 2.5. Comparative Genomic Hybridisation/Single-Nucleotide Polymorphism Analysis

Microarray analyses (array–comparative genomic hybridisation/single-nucleotide polymorphism [aCGH/SNP]) were used to detect copy number alterations (CNA) and copy number neutral loss of heterozygosity (CN-LOH). A QIAamp DNA Mini Kit (Qiagen Inc., Hilden, Germany) was used to isolate genomic DNA (gDNA). The gDNA was hybridised onto SurePrint G3 Cancer CGH + SNP 4 × 180 K Microarray (Agilent Technologies, Santa Clara, CA, USA) or HumanCytoSNP-12 (v2.1) BeadChip array (Illumina, San Diego, CA, USA). The arrays were scanned with a SureScan Dx Microarray Scanner (Agilent Technologies) or BeadArray Reader (Illumina) and the data analysed using Agilent Cytogenomics software v.5.0.1.–5.1.1.15 (Agilent Technologies) or BlueFuse Multi software v3.1–4.2 (Illumina). Detection limit of the aCGH/SNP was 15–20% of cells with genomic aberrations in the sample. Resolution limit for the detection of CNA/CN-LOH was ~350 kb/~5 Mb. All procedures were performed according to manufacturers’ protocols.

### 2.6. Tumourigenicity Assay

The use of animals was approved by the Commission for Animal Welfare of the First Faculty of Medicine, Charles University and the Ministry of Education, Youth, and Sports of the Czech Republic according to the animal protection laws. Tumourigenicity of FAP^+^ mesenchymal cells was tested using an orthotopic xenograft model [27]. FAP^+^ mesenchymal cells (0.5 × 10^6^ in 5 µL) were orthotopically implanted into 6–10-week-old male NOD.129S7(B6)-Rag1tm1Mom/J mice (The Jackson Laboratory, Bar Harbor, ME, USA) with a Hamilton syringe using a stereotactic device (Stoelting Co., Wood Dale, IL, USA, 1.2  mm anterior from bregma, 2.5  mm lateral from the midline, depth 3 mm). For two cell cultures, tumorigenicity was evaluated after 85 and 88 days (a typical interval after which tumor formation can be observed with patient-derived glioblastoma cells in our model). In the remaining two experiments, the interval was extended to 109 and 140 days, respectively, to reveal possible delayed tumor formation. The mice exhibited no clinical signs of intracranial tumors at the time of sacrifice. Brains were harvested, embedded in Tissue freezing medium (Leica, Allendale, NJ, USA), and coronal 10 µm sections were cut at various positions and stained using Differential Quik Stain Kit (Polysciences, Warrington, PA, USA). Since no tumor mass was found, sections from the area of implantation, which could be reliably identified by residual scaring of the tissue, were analysed using immunofluorescence staining of human nuclei (HuNu, clone 3E1.3, Merck, 1:500, overnight, 4 °C) and haematoxylin and eosin staining.

### 2.7. Transwell Migration Assay

A transwell migration assay was performed as described previously [27]. Cells (6 × 10^4^) were seeded in cell culture inserts with 8 μm pores (Corning, Glendale, AZ, USA) in an appropriate serum-free medium and allowed to migrate towards either the control or conditioned medium for 24 h. In experiments with FAP^+^ mesenchymal cells, media in the lower compartment contained 1% FCS, while glioma and endothelial cells were assayed under serum-free conditions. Non-migrated cells were removed using a cotton swab; cells on the lower surface of the insert were fixed with 5% glutaraldehyde in PBS and stained with methylene blue. Five microscopic fields per insert were photographed (20× objective) and cells counted manually.

### 2.8. Cell Growth Assays

For a direct co-culture, enhanced green fluorescent protein transfected U251 (U251pEGFP) were seeded at a density of 1.5 × 10^4^ cells/well in a 24-well plate alone or together with 1.5 × 10^4^ (1:1) or 0.75 × 10^4^ (2:1) FAP^+^ mesenchymal cells and allowed to grow in PM supplemented with 2% FBS, PGS, and P/S (ScienCell Research Laboratories). After 48 h, cells were harvested and counted on Coulter Counter Z2 (Beckman Coulter Czech Republic, Prague, Czech Republic). The percentage of U251pEGFP cells was then determined by flow cytometry using BD FACS Verse cytometer (BD Biosciences) and BD FACSuite Software (BD Biosciences) for evaluation.

For direct co-culture with HUVEC, subconfluent FAP^+^ mesenchymal cells or HBVP were pre-treated with 20 µg/mL mitomycin C (Merck) in serum-free PM for 2 h, washed twice with PBS, harvested, and seeded in M200 medium supplemented with 2% LVES at a density of 2 × 10^4^ cells/well in a 24-well plate. After 24 h, 6 × 10^4^ HUVEC were added into the wells. Wells with 2 × 10^4^ mitomycin C pre-treated mesenchymal cells and 6 × 10^4^ HUVEC without a co-culture were used as controls. M200 medium supplemented with 2% LVES was changed every 48 h. After 120 h, cells were harvested and counted on Coulter Counter Z2. The number of HUVEC was estimated by subtracting the number of mesenchymal cells from the total number of cells. 

To evaluate the effect of conditioned media on glioma cell growth, 0.5 × 10^4^ U251 and U87 cells were seeded in 1% FCS supplemented non-conditioned medium or conditioned media in 96-well plates. After 48 h, cell growth was evaluated using a CellTiter-Glo^®^ Luminescent Cell Viability Assay (Promega, Madison, WI, USA) according to manufacturer’s instructions. Luminescence signal was measured on a multilabel plate reader (Victor, Perkin Elmer, PE Systems, Prague, Czech Republic).

### 2.9. Immunohistochemistry and Immunocytochemistry, Quantification of FAP Expression in Glioblastoma 

FAP immunopositivity in tumour samples was detected in 4 µm paraffin sections using a primary rabbit monoclonal antibody (ab207178-clone EPR20021, Abcam, Cambridge, UK, 1:250, 20 min, room temperature (RT)) after antigen retrieval, which was performed with an EDTA-based pH 9.0 epitope retrieval solution (BOND Epitope Retrieval Solution 2) for 20 min. Bond Polymer Refine Detection (Leica) was used for visualisation of the primary antibody and haematoxylin counterstaining. Images were captured by an experienced pathologist (P.H.) on an Axioskop 2 mot plus microscope using the Axiocam ICc1 camera (Zeiss, Jena, Germany). FAP expression in cancer and stromal cells, vascularisation, and the presence of necrosis and microvascular proliferations were evaluated in five independent visual fields (10× objective). Vascularisation was scored on a four-tiered scale (0—no blood vessels, 1—sporadic blood vessels, 2—blood vessels in a substantial part of the visual field, 3—blood vessels in the entire visual field). FAP immunopositivity was assessed irrespective of its intensity using a four-tiered semiquantitative scale (Table 2 and Figure 1A). Average values for vascularisation and FAP expression in each sample were used for further analyses. Presence of necrosis and microvascular proliferations was scored as the percentage of visual fields containing these histomorphological hallmarks of GBM.

Double immunofluorescence staining of CD105 and FAP was performed in 10 µm frozen sections. Following fixation with 4% paraformaldehyde (10 min, RT), permeabilisation using 0.1% Triton X-100 (5 min, RT) and blocking with 10% FBS and 1% bovine serum albumin in Tris-buffered saline (TBS, 60 min, RT), the samples were sequential stained by an anti-FAP (clone D8, Applieddnasciences, Stony Brook, NY, USA, 1:100, overnight, 4 °C) and anti-CD105 antibody (clone SN6h, Dako AgilentSanta Clara, CA, USA, 1:200, 60 min, RT). The primary antibodies were visualised using anti-rat Alexafluor 488 (A-21208, ThermoFisher Scientific) and anti-mouse Alexafluor 546 (A-21202, ThermoFisher Scientific) labelled secondary antibodies (both 1:500, 60 min, RT). 

For immunocytochemistry, FAP^+^ mesenchymal cells and primary endothelial cells were grown on fibronectin-coated glass coverslips and HBVP on polylysine-coated glass coverslips. The cells were fixed with 4% paraformaldehyde (10 min, RT), permeabilised using 0.1% Triton X-100 (5 min, RT) and blocked with 10% FBS and 1% bovine serum albumin in TBS (60 min, RT). The following primary antibodies were used: anti-TE-7 (clone CBL271, Merck, 1:100, overnight, 4 °C), anti-α smooth muscle actin (αSMA, clone 1A4, Abcam, 1:200, overnight, 4 °C), anti-platelet derived growth factor receptor-β (PDGFRβ, clone PR7212, R&D systems, UK, 1:50, overnight, 4 °C), anti-FAP (mouse hybridoma F19, ATCC, 55 µg/mL, overnight, 4 °C), anti-von Willebrand factor (vWf, Dako Agilent, 1:200, 60 min, RT) and anti-glial acidic fibrillary protein (GFAP, clone GP-01, Abcam, 1:200, 60 min, RT). The primary antibodies were visualised using the corresponding secondary antibodies (anti-mouse Alexafluor 488, A-11001 and anti-rabbit Alexafluor 488, A-11008, ThermoFisher Scientific, 1:500, 60 min, RT). Hoechst 33258 (Merck, 50 ng/mL) added during the incubation with secondary antibodies was used for nuclear counterstaining. 

Samples were mounted in Aqua Poly/Mount (Polysciences, Hirschberg, Germany) and viewed and photographed using Olympus IX70 microscope equipped with a DP70BW camera (Olympus Czech Group, Prague, Czech Republic).

### 2.10. Proteome Profiler Human Angiogenesis Array

Conditioned media from FAP^+^ mesenchymal cells and HBVP were analysed using Proteome profiler human angiogenesis array (ARY007, R&D Systems) according to manufacturer’s protocol. The signal was detected using Chemidoc (BioRad, Prague, Czech Republic) with subsequent image analysis using Image Lab software (BioRad).

### 2.11. Chicken Chorioallantoic Membrane Assay (CAM Assay)

Fertilised chicken eggs (breed Brown Leghorn) were provided by the Institute of Molecular Genetics of the Academy of Science of the Czech Republic. Embryo development was initiated (day 0) by placing the eggs in the Nuve EN-400 incubator (37.5 °C, relative humidity 60%, KRD, Prague, Czech Republic). At day 3, we removed 3 mL of albumen though a small hole in the shell using a sterile syringe with a 21G needle. The hole was subsequently covered with an adhesive tape. At day 7, an observation window was created and eggs containing viable embryos selected for experiments. At day 10, a mixture of U87 cells (0.75 × 10^6^) with either FAP^+^ mesenchymal cells (0.25 × 10^6^) or HBVP (0.25 × 10^6^) in 10 μL of growth factor reduced LDEV-Free hESC-qualified Geltrex (Thermo Fisher, cat.no. A1413302) was implanted onto the chorioallantoic membrane. U87 cells (1 × 10^6^ in Geltrex) were used as a control. Tumours were allowed to grow for 5 days; they were photographed at days 11, 13, and 15. Experiments were terminated at day 15 by sacrificing the embryos and harvesting the tumours.

To evaluate the vascularisation of the tumours, 10 μm thick frozen sections were stained with haematoxylin and eosin. The entire area of the tumour was photographed using a 40× objective; areas containing haemorrhages and chorioallantoic membrane without tumour were excluded. Nucleated avian erythrocytes were manually counted by a blinded observer using ImageJ software (National Institutes of Health, Bethesda, MD, USA). At least 10 visual fields were evaluated for each tumour. 

### 2.12. 3D Angiogenic Sprouting Assay

3D angiogenic sprouting assay was performed according to Heiss et al. [28] with small modifications. In brief: HUVEC spheroids (800 cells/spheroid) were prepared using MicroTissues 3D petri dishes (256 recesses, Bio-port, Prague, Czech Republic) according to manufacturers’ instructions. After 24 h, spheroids formed in an agarose mould were harvested (approximately 256 spheroids), centrifuged (170× *g*, 5 min, RT), and carefully resuspended in 105 µL of FCS. The resuspended spheroids (105 µL) were gently mixed with 945 µL of a solution containing rat tail collagen I (420 µL; BD Bioscience), NaOH (9.66 µL; Merck), Methylcellulose (210 µL; Merck), and 305.34 µL of non-conditioned or FAP^+^ mesenchymal cell or HBVP-conditioned media. Concentration of collagen I, methylcellulose, and NaOH in the final suspension was 1.5 mg/mL, 20% and 9 mmol/L, respectively. The resulting suspension of spheroids was transferred into 24-well plates (500 μL per well). After 30 min of gelling at 37 °C in a tissue culture incubator, the solidified gels were carefully overlaid with 300 µL non-conditioned serum-free medium or conditioned media from FAP^+^ mesenchymal cells or HBVP. After 24 h, the spheroids were photographed and sprouts manually counted in a blinded manner to avoid bias using the ImageJ software. The experiments were performed in duplicates with at least 10–20 spheroids evaluated in each replicate. 

### 2.13. Statistical Analyses

Statistical analyses were performed with STATISTICA 12 (Tibco Software, Palo Alto, CA, USA). Shapiro-Wilk test, normal p-plots and Levene’s test were used to verify the assumptions for performing parametric tests. A two-sided *p* < 0.05 was considered statistically significant. 

## 3. Results

### 3.1. The Presence of FAP^+^ Stromal Cells Is Associated with Neovascularisation in Glioblastoma

Using immunohistochemistry in paraffin sections, we analysed FAP expression and localisation in 94 newly diagnosed GBMs and correlated it with vascularisation, presence of necroses, and proliferation activity. We observed variable FAP immunopositivity in individual cancer cells scattered in the tumour tissue or forming small clusters in 25.5% of tumours. FAP^+^ stromal cells, on the other hand, were present in 87.2% of tumours. They were represented mainly by perivascular spindle-shaped cells, which in some tumours extended into the parenchyma and formed FAP^+^ trabeculae of connective stroma (Figure 1A). FAP-expressing endothelial cells were detected rarely. This staining pattern is in line with the results of previous smaller studies showing FAP expression in glioma cells and in the stroma [8,9,13,29].

The quantity of FAP^+^ stromal cells positively correlated with tumour vascularisation (Kendal correlation coefficient 0.44, *p* < 0.001) and with the presence of microvascular proliferations (Kendal correlation coefficient 0.47, *p* < 0.001) (Figure 1B). We did not find any significant association between FAP^+^ stromal cells and either proliferation activity (Ki-67 labelling index) or the presence of necroses in GBM. We also found no significant correlation between the expression of FAP in cancer cells and tumor vascularization, microvascular proliferations, presence of necroses or Ki-67 labelling index. 

Interestingly, the blood vessels surrounded by FAP^+^ stromal cells were often dysmorphic, tortuous, and dilated. Moreover, in nearly 100% of cases their endothelial cells expressed CD105 (endoglin), a type III transforming growth factor (TGF) receptor and marker of activated endothelial cells in tumour-induced neovascularisation [30] (Figure 1C,D). Taken together, these data suggest that the FAP-expressing subpopulation of mesenchymal stromal cells in GBMs may be involved in the formation of new blood vessels and thus enhance tumour vascularisation. 

### 3.2. FAP^+^ Stromal Cells from Human Glioblastomas Express Mesenchymal Markers, Are Non-Tumorigenic, and Mostly Lack Aberrations Characteristic of Glioma Cells 

To determine the contribution of FAP^+^ stromal cells to GBM angiogenesis, we isolated this subpopulation from human GBMs by magnetic cell sorting using a specific anti-FAP antibody. We were able to effectively propagate FAP^+^ cells in culture conditions used for human brain vascular pericytes (HBVP); on the other hand, we did not succeed in maintaining them in DMEM supplemented with 10% FCS, which is traditionally used for culturing fibroblasts and glioma cells. We derived 28 primary cell cultures from 21 different human GBMs, which were subsequently characterised by immunocytochemistry. In addition to FAP, these cell cultures also expressed other mesenchymal markers–including TE-7 (100% of cultures), αSMA (59% of cultures), and PDGFRβ (88% of cultures)–consistent with the mesenchymal phenotype of FAP^+^ perivascular cells in human GBMs [8]. Sporadic immunopositivity (under 2% of stained cells) was observed for endothelial (vWf) and glial (GFAP) markers in 19% and 4% of the FAP-expressing cell cultures, respectively (Figure 2A,B, Appendix A).

We used aCGH/SNP to evaluate whether genetic alterations characteristic of GBMs are present in the isolated FAP-expressing cell cultures, i.e., whether these cells represent normal stromal cells recruited into the tumours or arise instead by transdifferentiation of transformed cells. Ten FAP^+^ cell cultures from different patients were compared to the corresponding tumour tissue from which they were derived. Alterations characteristic of GBMs were observed in nine out of ten tumour tissues (chromosome 7 gain/EGFR gene amplification *n* = 9, chromosome 10 loss including PTEN gene *n* = 8, Appendix A). In one tumour tissue, cytogenomic analysis revealed CN-LOH 17p and IDH1 gene mutation without typical GBM changes. On the other hand, we detected in a glioma stem-like cell culture derived from this tumour aberrations characteristic of GBMs (chromosome 7 gain, monosomy 10), which points to a secondary GBM. In contrast, the hallmarks of GBMs were absent in the vast majority of FAP^+^ cell cultures (Figure 2C, Appendix A). Normal karyotype was observed in three cultures (46A, 47A, 49A). Triploidy/tetraploidy without unbalanced structural aberrations was identified in culture 38A, four cultures (39B, 40, 42, 50A) contained changes which were not found in the corresponding GBM tissues. The most commonly observed aberrations in these cultures were gains/losses of gonosomes (Appendix A). Based on the mesenchymal phenotype and absence of mutations characteristic of GBMs, we assume that these cultures are most likely comprised of recruited stromal cells which acquired random alterations due to microenvironmental stress in the tumours or during in vitro propagation [31,32]. The remaining two FAP-expressing cultures (27A and 44A) were genetically highly similar to the original tissue (Figure 2C, Appendix A). This is in accordance with previous reports, which demonstrated the ability of glioma stem-like cells to give rise to cells phenotypically highly similar to endothelial [33], pericytic [34], and mesenchymal stem cells [35].

We further tested the tumourigenicity of FAP^+^ cell cultures. Orthotopic implantation of 0.5 × 10^6^ cells from four different FAP^+^ cell cultures in immunodeficient mice (2–3 per each culture) did not lead to neurological symptoms and tumour formation in any of the animals after a median follow-up of 98.5 days (range 85–140). Immunohistochemistry to detect human nuclei was used to more sensitively reveal the presence of implanted cells. For three cultures, no human cells were found. For one culture, individual isolated human cells could be identified in the immediate vicinity of the implantation site after 109 days without spreading into the surrounding tissue or forming a tumor mass.

Taken together, we successfully isolated FAP^+^ cells from human GBMs that express mesenchymal markers, are non-tumorigenic, and in most cases lack genetic aberrations characteristic of glioma cells, which suggests that they are non-malignant mesenchymal cells.

### 3.3. Endothelial Cells Attract FAP^+^ Mesenchymal Cells by Soluble Factors

Further, we tried to investigate whether the perivascular localisation of FAP^+^ stromal cells in GBM tissues may be the consequence of their attraction by endothelial cells. We tested the chemotactic effect of endothelial cell-conditioned media on the migration of five different FAP^+^ mesenchymal cell cultures with verified expression of αSMA, TE-7, and PDGFRβ. Although the basal migration of FAP^+^ cell cultures was highly variable, it was in all cases statistically significantly increased by HUVEC-conditioned medium (2.4–12.8-fold increase, Figure 3). These data suggest that endothelial cells have a potential to navigate FAP^+^ mesenchymal cells towards blood vessels. 

### 3.4. The Effect of FAP^+^ Mesenchymal Cells on the Migration and Growth of Endothelial and Glioma Cells

To elucidate the role of FAP^+^ stromal cells in the perivascular microenvironment of GBMs, we tested whether FAP^+^ mesenchymal cell cultures affect endothelial and glioma cells by secreted mediators and by direct cell-to-cell interactions. 

Using conditioned media from three different FAP^+^ mesenchymal cell cultures, we observed a statistically significant 1.9–8.4-fold increase of endothelial cell migration in a modified Boyden chamber assay (Figure 4A). Similarly, conditioned media from human brain vascular pericytes (HBVP) induced chemotaxis of endothelial cells (Figure 4A). On the other hand, none of these conditioned media promoted the growth of HUVEC [36]. A direct co-culture system allowing cell-to-cell contacts was then used to evaluate the effect on endothelial cell growth. Endothelial cells were seeded alone or in co-culture in 3:1 ratio with mitomycin-pretreated, growth-arrested mesenchymal cells. While HBVP statistically significantly increased the growth of endothelial cells, the effect of FAP^+^ mesenchymal cell cultures from GBMs was variable and not statistically significantly different from control (Figure 4B).

We further tested the effect of FAP^+^ mesenchymal cells on glioma cells. As was the case with endothelial cells, conditioned media statistically significantly increased the migration of glioma cells (2.0–7.5-fold increase for U87 and 1.9–7.7-fold increase for U251). A similar effect was also observed with HBVP-conditioned media (Figure 5A). To assess the effect of FAP^+^ stromal cells on glioma cell growth, we co-cultured them with enhanced green fluorescence protein expressing glioma cells (U251eGFP). After 48 h, the cells were counted and the proportion of glioma cells estimated by flow cytometry. Compared to control, the growth of glioma cells in the presence of FAP^+^ mesenchymal cells was higher (Figure 5B). To distinguish whether the observed effect is due to soluble factors or caused by direct cell-to-cell contacts, U251 and U87 glioma cells were cultured in conditioned media from FAP^+^ stromal cells. In both cell lines, the conditioned media led to a moderate but consistent increase of cell growth (1.1–1.2-fold, Figure 5C).

These data indicate that FAP^+^ mesenchymal cells affect endothelial and glioma cells by paracrine communication. 

### 3.5. FAP^+^ Mesenchymal Cells Promote Endothelial Sprouting and Contribute to Vascular Destabilisation

We applied two different approaches to determine whether FAP^+^ mesenchymal cells promote the formation of blood vessels. Using an in vitro 3D angiogenic sprouting assay and conditioned media from HBVP and FAP^+^ mesenchymal cells, we observed increased HUVEC sprouting (Figure 6A). Moreover, in three out of four tested FAP^+^ mesenchymal cell cultures, this effect was enhanced when the conditioned media were prepared under hypoxic (1% O_2_) conditions. The influence of hypoxia on endothelial cell sprouting was more pronounced when the pro-angiogenic effect of FAP^+^ mesenchymal cells was relatively low under normoxic conditions (Figure 6B). We further tested the effect of FAP^+^ mesenchymal cells on microvascular endothelial cells derived from human GBMs and observed a statistically significant increase of angiogenic sprouting upon exposure to the conditioned media (Figure 6C). Taken together, these data demonstrate that FAP^+^ stromal cells as well as HBVP secrete soluble factor(s) which stimulate endothelial cell sprouting.

To complement these results, we used a chorioallantoic membrane assay as an in vivo angiogenesis model. U87 cells, either alone or mixed with FAP^+^ mesenchymal cell cultures or with HBVP, were xenografted on a chorioallantoic membrane and allowed to form tumours. The growth of the xenografts was accompanied by formation of new blood vessels around and inside the tumours as reported in other studies [37]. Additionally, in several tumours we observed haemorrhages and blood clots. Their presence was statistically significantly more frequent in tumours containing FAP^+^ mesenchymal cells (89%) compared to tumours containing HBVP (39%) or U87 alone (28%, Figure 7). Visualisation and quantification of blood vessels in the xenografted tumours proved difficult due to unavailability of specific antibodies that would recognise avian vasculature. We have therefore estimated the amount of blood vessels by counting erythrocytes in haematoxylin and eosin-stained tissue sections. Nucleated avian red blood cells were identifiable in vessel-like structures and their quantity was higher in tumours containing FAP-expressing mesenchymal cells than in tumours formed from U87 glioma cells alone (Figure 7). 

In summary, both assays corroborate a proangiogenic effect of FAP^+^ mesenchymal cells. Results in the chorioallantoic membrane further suggest that these cells may contribute to destabilisation of the newly formed vasculature.

### 3.6. Changed Proportion of Pro-Angiogenic, Anti-Angiogenic and Blood Vessel-Stabilising Mediators in FAP^+^ Mesenchymal Cells

We used a proteome profiler array to identify potential mediators of the pro-angiogenic effect of FAP^+^ mesenchymal cells. Mediators promoting angiogenesis, such as Ang-2, Endothelin-1, and insulin growth factor binding protein (IGFBP)-2, were detected in conditioned media from all tested FAP^+^ mesenchymal cells and HBVP. On the other hand, the canonical angiogenesis mediators VEGF and IL8 were only detectable in HBVP and in cell culture 27B (Figure 8), while TGFβ1, basic fibroblast growth factor (bFGF), and epidermal growth factor (EGF) were present in very low quantities [36]. Compared to HBVP, several molecules contributing to blood vessel stabilisation, such as Ang-1, endostatin-2, IGFBP-3, and vasohibin, were consistently downregulated in the conditioned media from FAP^+^ mesenchymal cells. 

The Ang-2/Ang-1 ratio, a biomarker associated with increased microvessel density and angiogenesis [38], was substantially higher in FAP^+^ mesenchymal cells (3.2–14.9 vs. 0.24 in HBVP, Figure 8) suggesting a possible mechanism of blood vessel destabilisation by these cells.

### 3.7. Presence of FAP^+^ Stromal Cells Is Associated with Tumour Progression 

We evaluated whether FAP expression in cancer and stromal cells as determined by immunohistochemistry is associated with tumour progression in patients with newly diagnosed GBM. In clinical trials, a frequently used parameter of GBM progression is progression-free survival at six months [39,40]. In our patient cohort (*n* = 90), 35.6% of patients showed no signs of progression six months after surgical removal of the tumour, while 64.4% of patients had progressed as determined by MRI or clinical deterioration. Using logistic regression, we identified two parameters which were associated with progression at this time interval: expression of FAP in stromal cells (*p* = 0.006, Table 3) and the extent of resection (odds ratio 5.1 for subtotal vs. radical resection, *p* = 0.010). FAP expression in cancer cells was not significantly associated with GBM progression at six months.

Cox regression analysis revealed an association between overall survival and the extent of resection (hazard ratio 2.9 for subtotal vs. radical resection, *p* = 0.0010), the type of administered therapy (hazard ratio 0.096 for Stupp protocol vs. radiotherapy alone or no adjuvant therapy, *p* < 0.0001) and age (hazard ratio 1.042, *p* = 0.0018). Except for age, these factors were also associated with progression-free survival (extent of resection: hazard ratio 2.05 for subtotal vs. radical resection, *p* = 0.0085, therapy: hazard ratio 0.44 for Stupp protocol vs. radiotherapy alone or no adjuvant therapy, *p* = 0.0140). The relationship between survival and adjuvant therapy is nevertheless complicated by the fact that patients with rapid progression or poor postoperative clinical status were not able to undergo the complete Stupp protocol. Histopathological parameters (Ki67, presence of necrosis, vascularity, or quantity of microvascular proliferations) and FAP expression in stromal or cancer cells were not significantly associated with progression-free and overall survival.

## 4. Discussion

FAP is a membrane-bound serine protease that is characteristically upregulated in numerous malignancies. Due to its cell surface localisation and selective overexpression in tumours, FAP is considered a potential diagnostic and therapeutic target and several approaches for its targeting were recently developed [41]. Although FAP can be expressed by malignant cells and is thought to contribute to their invasive properties [42], it is viewed primarily as a robust marker of non-malignant stromal cells, such as cancer-associated fibroblasts (CAFs). FAP^+^ CAFs maintain tumour-permissive microenvironment by several mechanisms and play a well-documented role in the pathogenesis of epithelial cancers [10]. CAF-like cells have been described in malignancies of the central nervous system [43,44] but their possible contribution to gliomagenesis remains largely unexplored.

In our previous work, we reported an upregulation of FAP in GBMs and identified a subpopulation of FAP^+^ stromal cells which expressed mesenchymal markers and were typically localised around blood vessels [8,45]. In the present study, we show that the blood vessels surrounded by FAP^+^ stromal cells are comprised of activated CD105^+^ endothelial cells and moreover, the quantity of FAP^+^ stromal cells positively correlates with vascularisation and GBM progression. Using in vitro and in vivo models, we further demonstrate that FAP^+^ mesenchymal cells induce endothelial cell migration and sprouting and stimulate the growth and migration of glioma cells by soluble mediators.

Rampant angiogenesis, which results from intrinsic changes in endothelial cells and their altered interactions with other constituents of the tumour microenvironment, is one of the hallmarks of GBMs. Our results indicate that blood vessels surrounded by FAP^+^ stromal cells are often dilated, tortuous, and dysmorphic. Endothelial cells of these vessels express CD105 (endoglin), an activation marker characteristically expressed in tumour-associated vasculature [30], which suggests ongoing neoangiogenesis. This is further supported by our observation that FAP expression in stromal perivascular cells positively correlates with semi-quantitatively estimated vascularisation and presence of microvascular proliferations in GBMs. These results are concordant with recent transcriptomic analyses which demonstrated upregulation of FAP gene expression in hyperplastic blood vessels and microvascular proliferations [13]. The role of FAP in angiogenesis is suggested also by other studies. FAP expression correlates with microvessel density in lung cancer [46] and is induced in stromal cells surrounding newly formed blood vessels in injured cornea [47]. Experimental work in animal models of breast [48] and colon [46] cancers further suggests that in both malignant and stromal cells, FAP may be directly involved in cancer angiogenesis. Precise mechanisms of these proangiogenic effects are the subject of ongoing debate. FAP cleaves neuropeptide Y and converts it into a pro-angiogenic Y2 receptor agonist [49]. Moreover, it has been shown that FAP enzymatic activity in fibroblasts increases the expression of Ang-1 and VEGF-C, although the underlying molecular mechanisms are yet to be described [20]. Other reports indicate that hepatic stellate cells, tissue-specific pericytes in the liver, overexpress FAP following activation [50] and promote angiogenesis in liver fibrosis and hepatocellular carcinoma by increasing the levels of Angiopoietin-1 and Angiopoietin-2 [51,52].

The perivascular niche in GBMs is populated by various cell types, such as endothelial cells, pericytes, mesenchymal and glioma cells, as well as immune cells, and is a key player in various aspects of GBM biology including neoangiogenesis and the spreading of glioma cells. To better understand the role of FAP^+^ stromal cells in the perivascular niche, we isolated these cells from GBM tissue and analysed their effect on endothelial and glioma cells. FAP^+^ cells were efficiently propagated in conditions suitable for pericytes, expressed mesenchymal markers PDGFRβ, αSMA, and TE-7, but were negative for glioma (GFAP) and endothelial (vWf) markers. This pattern of phenotypic markers corresponded to FAP^+^ stromal cells in GBM tissues [8] as well as to in vitro cultured human brain vascular pericytes (HBVP, [36]). FAP^+^ mesenchymal cells were not tumorigenic and genomic aberrations characteristic of GBMs were absent in 80% of tested cultures. This is consistent with our previous observation that perivascular FAP^+^ cells in GBM lack EGFR amplification [8] and suggests that they represent recruited non-malignant host cells. In contrast, 20% of FAP^+^ mesenchymal cell cultures in our study resembled original tumour tissues on the genomic level. Given their mesenchymal phenotype, these cells most likely represent transdifferentiated glioma stem-like cells, which have previously been reported to give rise to mesenchymal stem cells [35] and pericytes [34]. 

The phenotype of FAP^+^ mesenchymal cell cultures isolated from GBMs is similar to cancer-associated fibroblasts (CAF), which represent the main subpopulation of FAP^+^ stromal cells in carcinomas [53]. CAF-like cells have been previously isolated from brain tissue surrounding GBMs by Clavreul et al., but FAP expression was in that study not evaluated [43,54]. On the other hand, the perivascular localisation of FAP^+^ mesenchymal cells in GBM tissue suggests that they might represent a subpopulation of pericytes. Indeed, pericyte markers PDGFRβ and NG-2 (preliminary results, [36]) were expressed in FAP^+^ mesenchymal cell cultures and recent analyses using single-cell RNA sequencing and immunodetection in GBM demonstrated FAP expression in pericytes [13,55]. Endothelial cells attract pericytes during blood vessel formation [56], which corresponds with our observation that HUVEC-conditioned media chemoattract FAP^+^ mesenchymal cells. Microenvironmental cues can initiate and guide the process of pericyte differentiation into other mesenchymal cell types, including collagen type-I producing fibroblast-like cells [57] and CAFs [55,58], and pericytes are thus currently viewed as intermediates on the phenotypic continuum of mural cells [58,59].

Under physiological conditions, pericytes play an important role in several aspects of postnatal angiogenesis [60]. In addition to stabilising newly formed blood vessels, pericytes contribute to the early phase of vessel development. After angiogenic stimulation, pericytes localise at sprouting tips and bridge the gap between the tips of opposing endothelial sprouts before their fusion [61]. Our results demonstrate that in normoxic conditions, GBM-associated FAP^+^ mesenchymal cells increase endothelial cell migration and sprouting of both macrovascular (HUVEC) as well as microvascular endothelial cells. Exposure of FAP^+^ mesenchymal cells to hypoxia further enhances endothelial cell sprouting, especially in cell cultures where the proangiogenic effect is under normoxic conditions relatively modest. In our experiments, human brain vascular pericytes (HBVP)—which were used as a physiological control–induced endothelial cell migration and sprouting. They also promoted endothelial cells growth, which contrasts with previously reported results [62]. It is important to note that the commercially available pericytes were derived from human foetal tissues (personal communication with the provider). In the developing human brain, pericytes play a crucial and unique role in angiogenesis by guiding endothelial cells during vessel growth [63,64], which may explain their pro-proliferative effect in our in vitro experiments.

In the chorioallantoic membrane GBM model, tumours containing FAP^+^ mesenchymal cells were more vascularised and developed haemorrhages more frequently than either glioma cells alone or glioma cells with admixture of human brain vascular pericytes (HBVP). We speculate that the haemorrhages may be the result of increased formation of immature blood vessels and their insufficient stabilisation. This hypothesis is supported by previous observation according to which increased haemorrhage in human GBM specimens is linked to increased angiogenesis [33] 

Conditioned media derived from FAP^+^ mesenchymal cells and human brain vascular pericytes contained detectable levels of several pro-angiogenic mediators, such as Ang-2, Endothelin-1, and IGFBP-2, but there was no consistent difference between the two cell types. Nevertheless, the levels of various anti-angiogenic and vessel-stabilising mediators—such as Ang-1, Endostatin-2, IGFBP-3, and Vasohibin—were substantially lower in FAP^+^ mesenchymal cells compared to human brain vascular pericytes. In the context of blood vessel stability, the disbalance between Ang-1 and Ang-2 in FAP^+^ mesenchymal cell-conditioned media seems to be of particular relevance. RNA in situ hybridisation demonstrated that Ang-1 and VEGF mRNA are expressed in pericyte-like perivascular mural cells, while Ang-2 mRNA was detected in both the pericyte-like and endothelial cells [65]. Ang-1 promotes endothelial cell survival and blood vessel maturation [66], while Ang-2 acts as Ang-1 antagonist [3]. It has been proposed that Ang-1 restricts the effect of VEGF on angiogenesis [67]. On the other hand, Ang-2 cooperates with VEGF and in its presence promotes angiogenesis, but also leads to vessel destabilisation when VEGF levels are low [4]. According to our data, VEGF expression was lower compared to HBVP in two out of three tested FAP^+^ stromal cells. This could intensify the effect of Ang-1/Ang-2 disbalance and contribute to the development of haemorrhages in our model. A recent study demonstrated that in a rat glioma model, pericytes not only increased vessel formation but also caused their destabilisation, but the mechanism responsible for this effect is yet to be identified [68]. 

It has been suggested that glioma cells are attracted into the perivascular niche and their proliferation is enhanced at vascular branch points, which may be mediated by soluble factors released by endothelial cells [15,16,17,69]. To the best of our knowledge, the effect of pericytes on glioma cell migration or proliferation is largely unexplored. Our experiments suggest that perivascularly localised FAP^+^ mesenchymal cells may contribute to the vascular tropism of glioma cells and enhance their proliferation. Conditioned media from FAP^+^ mesenchymal cells and HBVP promoted chemotaxis of glioma cells. In addition, glioma cell growth was modestly increased in a direct co-culture as well as under the influence of soluble factors released by FAP^+^ mesenchymal cells. These results complement earlier studies by Clavreul et al. [42,70] who isolated CAF-like stromal cells from the peritumoral zone of GBMs and showed that part of these cultures promoted glioma cell growth. Collectively, these studies support the notion that various mesenchymal cell types in the GBM microenvironment, including a subpopulation of FAP^+^ mesenchymal cells, may facilitate growth and characteristic perivascular spreading of glioma cells.

A meta-analysis of 15 studies in solid tumours concluded that FAP is a negative prognostic factor in several human malignancies, especially when it is expressed in cancer cells [71]. The data in GBMs are, however, inconsistent. Based on the TCGA data, Ebert et al. reported that tumours with the highest FAP expression (top 10%) had worse prognosis than tumours with the lowest FAP expression (bottom 10%) [13] but our previous analysis of primary GBMs in the TCGA dataset did not reveal an association with survival when quartiles were used as a cut-off. Moreover, the overall quantity of FAP as determined by ELISA or qRT-PCR was not associated with overall survival in our cohort of 43 GBM patients [8]. The design of these analyses did not, however, enable to address the possible prognostic role of FAP expression in individual cell types in the tumour microenvironment. In this, so far the largest cohort of patients reported in literature, we therefore quantified FAP in cancer and stromal cells separately and analysed their association with survival. In agreement with previous smaller studies [8,13], FAP expression was detected in stromal cells in a large proportion of patients. On the other hand, FAP expression in cancer cells was much less frequent and most GBMs contained no or few FAP^+^ cancer cells. We observed that the expression of FAP in stromal cells, but not in cancer cells, was associated with GBM progression at six months after surgery. Nevertheless, we found no statistically significant association between the expression of FAP or other histopathological parameters and overall or progression-free survival, which suggests that in our patient cohort these outcome measures were influenced mainly by clinical variables, type of administered therapy, and response to it. We speculate that FAP^+^ stromal cells may play a role especially in the initial stage of tumour recurrence by several ways: (1) we demonstrated that FAP^+^ mesenchymal cells have a moderate but consistent effect on the proliferation of glioma cells, which is at least in part mediated by soluble factors; (2) at the same time, they increase glioma cell migration and due to their perivascular localisation may navigate glioma cells to one of the major pathways through which glioma cells spread in vivo; (3) finally, FAP^+^ mesenchymal cells can promote angiogenesis and according to some studies [72], vascularisation may be linked to worse prognosis in patients with malignant gliomas.

A possible limitation of our study might be that due to the limited number of cells available for a particular patient derived cell culture, some experiments could be performed only once. We have tried to mitigate this limitation by performing the assays for several independent cell cultures isolated from different glioblastoma patients and when more material was available, we preferred to use the particular cell culture for additional analyses. In addition, when experiments were repeated for some of the cultures, the results were reproducible.

## 5. Conclusions

We isolated and characterised FAP^+^ mesenchymal cells from human GBMs that share some characteristics with cancer-associated pericytes and/or fibroblasts and facilitate GBM progression due to their potential to promote neoangiogenesis and enhance glioma cell growth and migration by paracrine interactions.

## Figures and Tables

**Figure 1 cancers-13-03304-f001:**
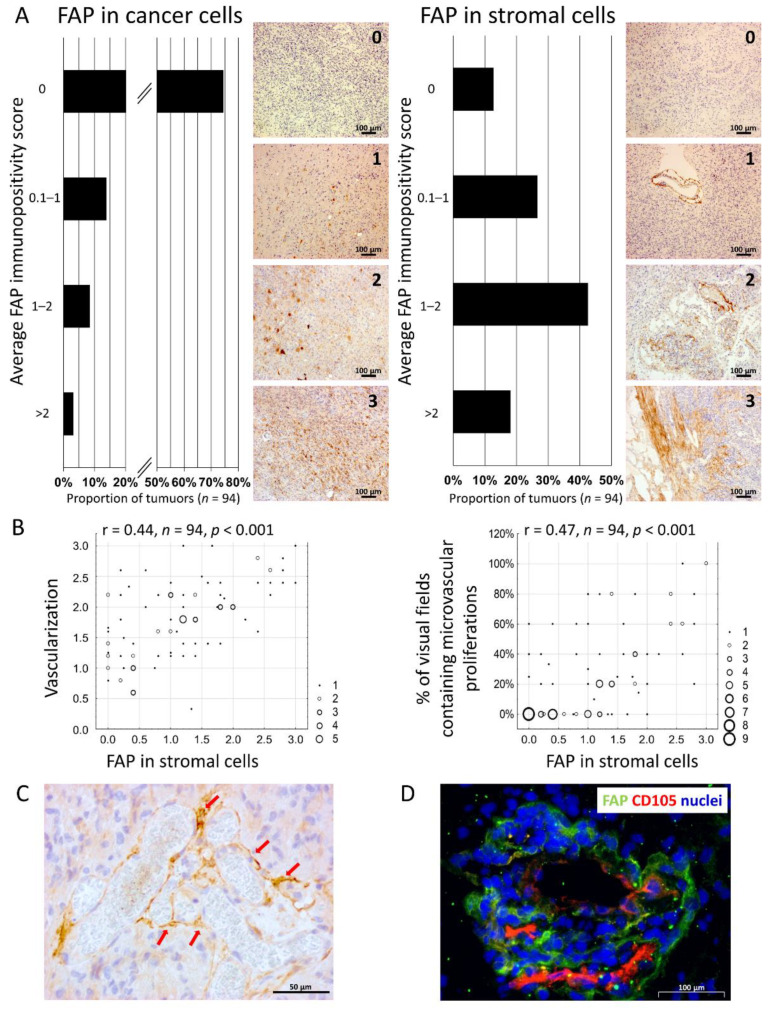
The quantity of FAP-expressing stromal cells correlates with vascularisation in glioblastoma. (**A**) Summary of FAP expression in cancer and stromal cells in glioblastomas evaluated by immunohistochemistry (*n* = 94, average immunopositivity score for each tumour was calculated from five independent microscopic fields, depicted are representative images illustrating the four-tiered semiquantitative scale used to evaluate FAP expression). (**B**) Correlation between FAP expression in stromal cells, vascularisation, and the presence of microvascular proliferations. Frequencies of overlapping points between two variables are illustrated by the size of the point markers. Kendal correlation coefficient was used for the statistical evaluation of the data. (**C**) A representative image of dysmorphic, tortuous, and hyperplastic blood vessels surrounded by FAP-expressing cells (red arrows). (**D**) A representative image of CD105^+^ endothelial cells surrounded by FAP-expressing stromal cells. Nuclei were counterstained with Hoechst.

**Figure 2 cancers-13-03304-f002:**
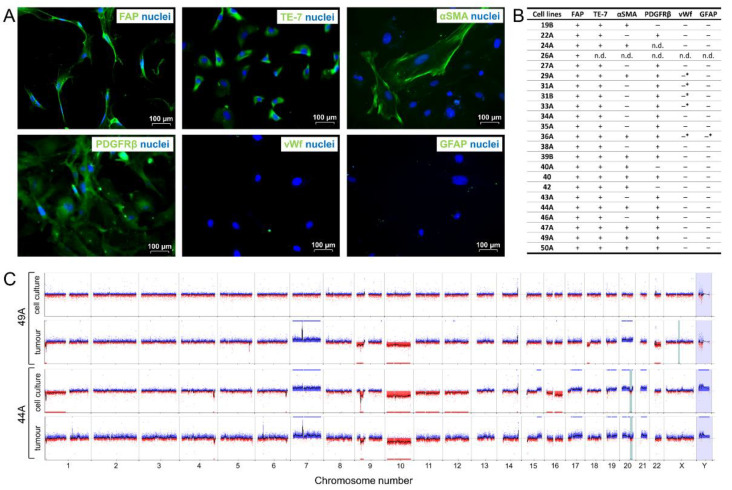
FAP^+^ stromal cell cultures derived from human glioblastoma characteristically express mesenchymal markers and in most cases lack the genetic aberrations characteristic of glioma cells. (**A**) Immunocytochemical detection of mesenchymal (TE-7, αSMA, PDGFRβ), endothelial (vWf), and glial (GFAP) markers in FAP^+^ mesenchymal cell cultures. (**B**) Summary of immunocytochemical analyses in FAP^+^ cell cultures used for functional studies in this project; (+) positivity in most cells with the exception of αSMA, where only a subpopulation of cells was positive, (–) no positive cells detected. (–*) negativity in most cells with admixture of less than 2% of positive cells. (**C**) aCGH/SNP analysis of FAP^+^ mesenchymal cell cultures and the corresponding tumour tissues. Cell culture 49A represents the dominant group of FAP^+^ mesenchymal cell cultures without changes characteristic of glioblastomas, whereas 44A typifies the smaller group that bears aberrations similar to the corresponding tumour tissue. Losses of DNA copy number in red, gains of DNA copy number in blue. CN-LOH is indicated by a light blue bar.

**Figure 3 cancers-13-03304-f003:**
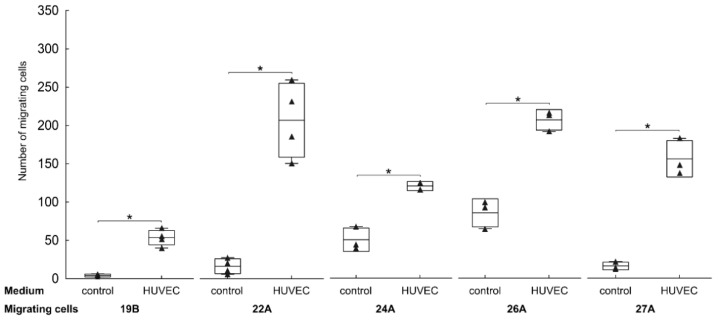
FAP^+^ mesenchymal cells are attracted by endothelial cells in vitro. Chemotaxis of FAP^+^ mesenchymal cell cultures (19B, 22A, 24A, 26A, 27A) towards control (non-conditioned) medium and to human umbilical vein endothelial cell (HUVEC)-conditioned medium. The media were supplemented with 1% foetal calf serum. Cell migration was evaluated using a modified Boyden transwell migration assay after 24 h. Results are presented as average number of migrating cells, for cell culture 19B one representative experiment of two is shown, for 22A, 24A, 26A and 27A one experiment was performed in at least triplicates depending on cell availability. * *p* < 0.05 compared to the corresponding control, Welsch’s *t*-test. Line—mean, box—mean ± SD, whiskers—range, triangles—raw data.

**Figure 4 cancers-13-03304-f004:**
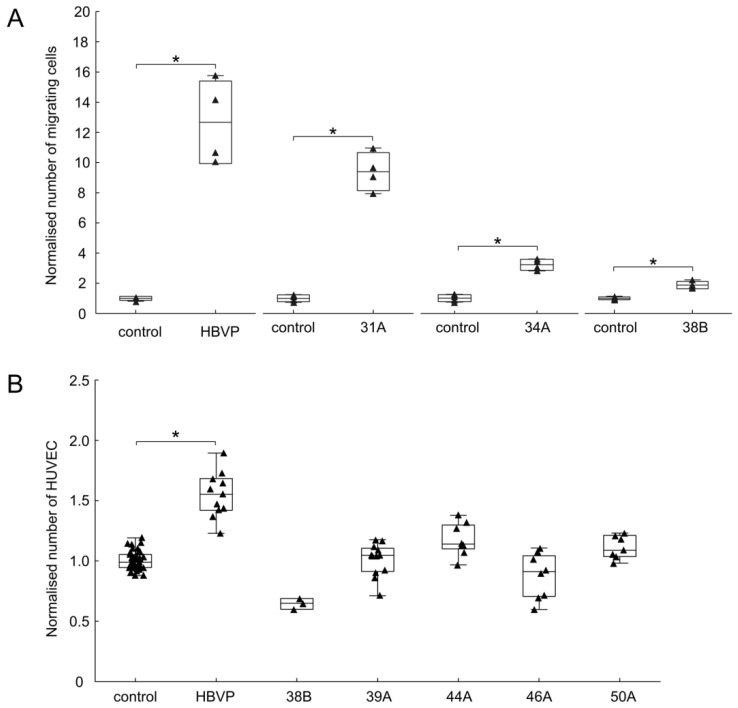
FAP^+^ mesenchymal cells enhance the migration of endothelial cells. (**A**) Chemotaxis of human umbilical vein endothelial cell (HUVEC) towards control (non-conditioned) medium and towards conditioned media from FAP^+^ mesenchymal cells (31A, 34A, 38B) and human brain vascular pericytes (HBVP), respectively. Cell migration was evaluated using a modified Boyden transwell migration assay after 24 h. Data in each experiment were normalised to control. One experiment for each conditioned medium from a particular cell culture performed in quadruplicates. Line—mean, box—mean ± SD, whiskers—range, triangles—raw data. * *p* < 0.05 compared to the corresponding control, Welsch’s *t*-test. (**B**) Effect of FAP^+^ mesenchymal cells on the growth of HUVEC in a direct co-culture system. HUVEC were grown alone or mixed with mitomycin-pretreated HBVP or FAP^+^ stromal cells (38B, 39A, 44A, 46B, 50A), the number of HUVEC cells was evaluated after 120 h. Results from three (HBVP) and two (39A, 44A, 46A, 50A) independent experiments performed in quadruplicates, and one (38B) experiment in triplicates. Data in each experiment were normalised to HUVEC grown alone. Line—median, box—25th to 75th percentile, whiskers—non-outlier range, triangles—raw data. * *p* < 0.05 compared to control, Kruskal-Wallis test.

**Figure 5 cancers-13-03304-f005:**
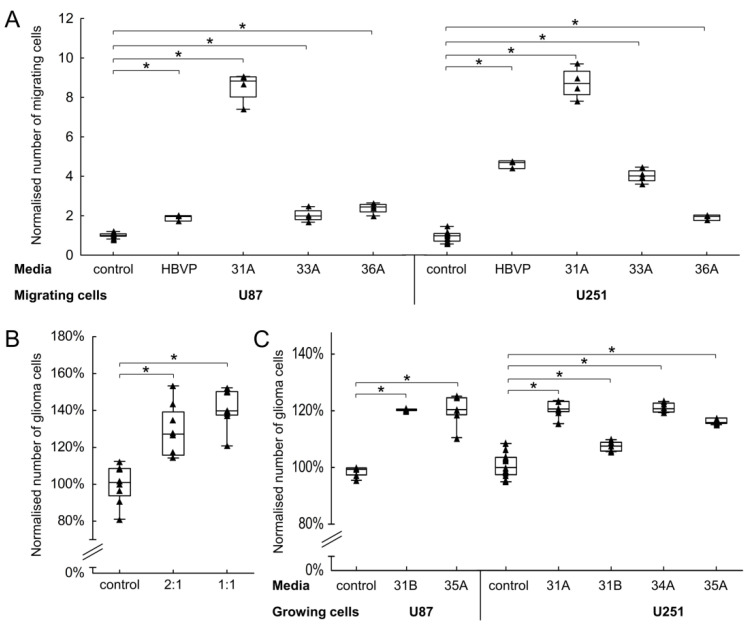
FAP^+^ mesenchymal cells increase the growth and migration of glioma cells. (**A**) Chemotaxis of glioma cells (U251, U87) towards non-conditioned medium and conditioned media from FAP^+^ mesenchymal cell cultures (31A, 33A, 36A) and from human brain vascular pericytes (HBVP), respectively. Cell migration was evaluated using a modified Boyden transwell migration assay after 24 h. Results from two independent experiments performed in quadruplicates, data in each experiment were normalised to control. (**B**) Effect of FAP^+^ mesenchymal cells on the growth of glioma cells in a direct co-culture system. Fluorescently labelled U251 glioma cells were grown alone or mixed with 38A FAP^+^ mesenchymal cells in ratios 2:1 and 1:1; the number of glioma cells was evaluated after 48 h. Results from two independent experiments performed in quadruplicates, data in each experiment were normalised to control. (**C**) Growth of glioma cells (U251, U87) in non-conditioned medium and conditioned media from FAP^+^ mesenchymal cell cultures (31A, 31B, 34A, 35A). All media were supplemented with 1% foetal calf serum; cell growth was evaluated after 48 h. Results from two independent experiments performed in quadruplicates, data in each experiment were normalised to control. Line—median, box–25th to 75th percentile, whiskers—non-outlier range, triangles—raw data. * *p* < 0.05 compared to control, ANOVA, Dunnett post hoc test.

**Figure 6 cancers-13-03304-f006:**
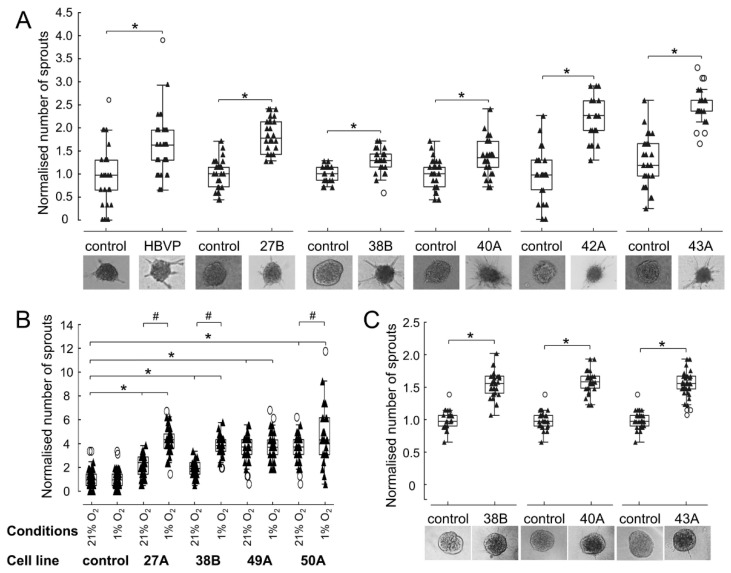
FAP^+^ mesenchymal cells increase angiogenic sprouting. (**A**) Human umbilical vein endothelial cell (HUVEC) sprouting in a non-conditioned serum-free medium and conditioned serum-free media from FAP^+^ mesenchymal cell cultures (27B, 38B, 40A, 42A, 43A) and human brain vascular pericytes (HBVP). One experiment for each conditioned medium from a particular cell culture. (**B**) Sprouting of HUVEC in a non-conditioned serum-free medium and conditioned serum-free media from FAP^+^ stroma cell (27B, 38B, 40A, 50A) prepared under normoxic (21% O_2_) or hypoxic (1% O_2_) conditions. Results from two independent experiments. (**C**) Primary endothelial microvascular cell sprouting in a non-conditioned serum-free medium and conditioned serum-free media from FAP^+^ mesenchymal cells (38B, 40A, 43A). Two experiments for each conditioned medium from a particular cell culture. Endothelial sprouting was evaluated by a 3D angiogenic sprouting assay after 24 h. Representative images of spheroids in the corresponding media are shown. Data were normalised to control, two independent wells each containing 10–20 spheroids were evaluated for control and conditioned media in each experiment. Line—median, box—25th to 75th percentile, whiskers—non-outlier range, triangles—raw data. * *p* < 0.05 compared to control, # *p* < 0.05 normoxic vs. hypoxic conditions. Mann—Whitney test (**A**,**C**), ANOVA, Tukey post hoc test (**B**).

**Figure 7 cancers-13-03304-f007:**
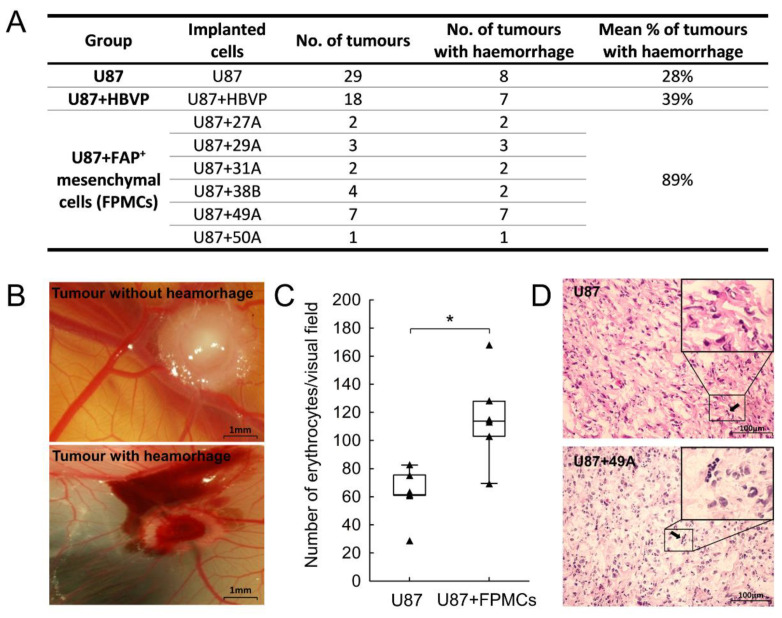
FAP^+^ mesenchymal cells contribute to the development of haemorrhages in a chorioallantoic membrane glioblastoma model. (**A**) The number of tumours with haemorrhages and their proportion in each group (U87 and U87 + HBVP and U87 + FAP^+^ mesenchymal cell cultures 27A, 29A, 31A, 38B, 49A, 50A). Pearson Chi-square *p* = 0.0001; Fisher’s exact two-tailed test for tumours containing FAP^+^ mesenchymal cells compared to tumours containing HBVP *p* = 0.0019 and compared to U87 alone *p* < 0.0001. (**B**) Representative images of tumours from U87 (without haemorrhage) and U87 mixed with FAP^+^ mesenchymal cells (with haemorrhage) growing on a chorioallantoic membrane for five days. (**C**) Quantification of erythrocytes in vessel-like structures in tumours from U87 and U87 with FAP^+^ mesenchymal cells (U87 + FPMCs), *n* = 7–8 tumours in each group. Line—median, box—25th to 75th percentile, whiskers—non-outlier range, triangles—raw data. * *p* < 0.05, Mann—Whitney test. (**D**) Representative haematoxylin and eosin-stained histology sections of tumours growing on chorioallantoic membrane; black arrow shows a vessel-like structure filled with nucleated avian erythrocytes in the tumour mass.

**Figure 8 cancers-13-03304-f008:**
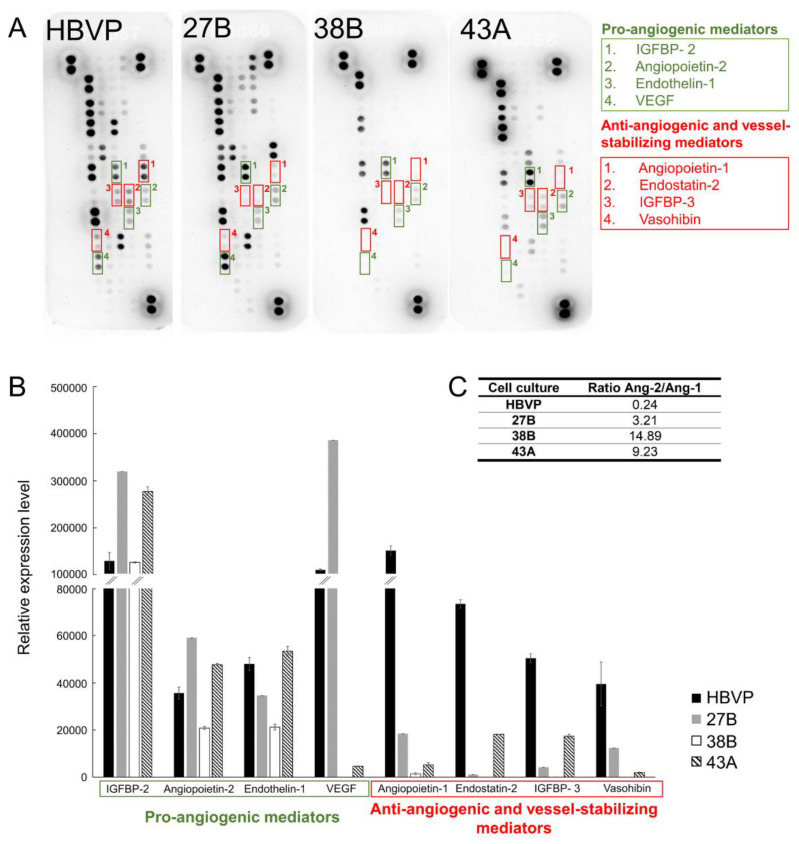
FAP^+^ mesenchymal cells from human glioblastomas produce mediators that regulate angiogenesis. (**A**) Detection of angiogenesis-related mediators by proteome profiler human angiogenesis array in conditioned media from FAP^+^ mesenchymal cells (27B, 38B, 43A) and human brain vascular pericytes (HBVP). (**B**) Densitometric analysis of selected pro-angiogenic, anti-angiogenic and blood vessel-stabilising molecules in conditioned media. Dot densities were quantified using Image Lab software (BioRad) and are presented as means ± SD. (**C**) Ratio between the relative levels of Ang-2 and Ang-1 in conditioned media.

**Table 1 cancers-13-03304-t001:** Characteristics of glioblastoma patients evaluated in the survival analysis.

Number of Patients	90
Age–median (range)	63.2 (28.6–81.5)
Sex	56M/34F
Extent of resection	62 radical, 28 subtotal
Preoperative Karnofsky performance status (median, range)	90 (40–100)
Overall survival–median (range)	13.6 months (3.1–57.9)
Progression-free survival–median (range)	5 months (2–39)

**Table 2 cancers-13-03304-t002:** Semiquantitative FAP immunopositivity scoring in glioblastoma sections (see Figure 1 for representative images).

Score	FAP in Cancer Cells	FAP in Stromal Cells
0	negative	negative
1	1–10% positive cells	perivascular positivity in sporadic blood vessels
2	10–20% positive cells	perivascular positivity in over 2/3 of the visual field
3	more than 20% positive cells	extensive perivascular positivity and FAP^+^ trabeculae

**Table 3 cancers-13-03304-t003:** FAP expression in stromal cells is associated with a higher probability of progression at six months in glioblastoma. Estimated odds ratios and 95% confidence intervals for the presence of tumour progression at six months for tumours with differing FAP immunopositivity in stromal cells.

FAP Immunopositivity in Stromal Cells	Odds Ratio	Odds Ratio 95% Confidence Interval
1 vs. 0	9.9	2.0–50.1
2 vs. 0	18.8	2.4–145.6
3 vs. 0	27.8	2.8–279.8
2 vs. 1	1.9	1.2–3.0
3 vs. 1	2.8	1.3–6.0
3 vs. 2	1.5	1.1–2.0

## Data Availability

The data related to this article are presented in the manuscript and Appendix A.

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
