# Peer review of "Fibroblast Activation Protein Expressing Mesenchymal Cells Promote Glioblastoma Angiogenesis"

_cancers, 2021, doi:10.3390/cancers13133304_

Round 1

Reviewer 1 Report

This work by Balaziova et al. reveals the significant role of FAP+ mesenchymal cells in GBM-associated angiogenesis. The authors demonstrate via in vitro and in vivo models that FAP+ mesenchymal cells promote not only angiogenesis but also glioma cell growth and migration, and are positively associated with GBM progression.

Overall, this is a well-written and organised paper. The findings of this paper shed further light into the role of the microenvironment in GBM biology and suggest that FAP+ mesenchymal cells may represent potential therapeutic targets for tackling GBM progression.

Author Response

We thank the reviewer for the positive  and stimulating evaluation of the manuscript.

Eva Balaziova, Aleksi Sedo, Petr Busek on behalf of the authors

Reviewer 2 Report

The authors present an interesting paper examining the role of FAP+ mesenchymal cells in the GBM microenvironment and have undertaken an extensive analysis demonstrating that FAP+ stromal cells may contribute to glioblastoma progression by promoting vascular angiogenesis and enhancing glioma cell migration and proliferation through the secretion of soluble factors.

Overall, this is a well-written paper with interesting results and well supported conclusions that help to expand our knowledge regarding the dynamic interaction of non-neoplastic stromal cells within the glioblastoma tumour microenvironment.

There are a few minor points that need addressing.

Introduction -Line 65-66: the statement should reference the original Nature publication identifying the involvement of p53, Rb and PI3K pathways in GBM progression: (TCGA) Cancer Genome Atlas Research Network. (2008). Comprehensive genomic characterization defines human glioblastoma genes and core pathways. Nature 455, 1061–1068

Methods 2.12: was the photographing of spheroids and the manual counting of sprouts performed in a blinded manner to avoid bias?

Results 3.1-Line 351-352: it is not clear what histopathological parameters are being referred to here.

Results 3.2-Lines 412-415: orthotopic implantation of the FAP+ stromal cells did not lead to tumour formation. In this experiment were euthanised after a median follow up time of 85-140 days which is a relatively broad time range. As tumour formation was not observed, the clinical implications of tumour formation were obviously not an indication for euthanasia however it is not clear from the methods what the criteria or clinical signs that prompted the implementation of euthanasia were and whether this was considered normal.

It is also not clear from the results what was observed when examining 10 uM sections stained with HuNu. Were the stromal cells originally implanted visible but showed no signs of growth or were the cells not evident at all? How many 10 uM sections were examined per mouse and how was it determined that what was observed did not constitute tumour formation? As there are no figures or data presented for this experiment, it is difficult to determine how thoroughly analysed and how convincing this data is.

Results 3.4-It is not clear that the assumptions for performing a Student T-test (Fig 4A, 5A) or ANOVA (Fig 4B, 5C) with regards to normality and homogeneity of variance have been met. Statistical analyses should confirm that the appropriate tests eg. Shapiro-Wilks, Levene’s etc) have been employed to confirm the assumptions are complied with.

Results 3.6/Figure 8: text and figures are inconsistently labelled as either 27A or 27B. Please correct.

Results 3.7-Line 601-602: it would be more accurate to state ‘we identified two parameters which correlate with progression’ than ‘we identified two parameters which influence progression’.

Author Response

We appreciate the reviewer’s constructive suggestions. We have made the corresponding changes in the revised manuscript.

Point by point response:

Introduction -Line 65-66: the statement should reference the original Nature publication identifying the involvement of p53, Rb and PI3K pathways in GBM progression: (TCGA) Cancer Genome Atlas Research Network. (2008). Comprehensive genomic characterization defines human glioblastoma genes and core pathways. Nature 455, 1061–1068

Response: The reference was changed as suggested by the reviewer. 

Methods 2.12: was the photographing of spheroids and the manual counting of sprouts performed in a blinded manner to avoid bias?

Response: We strived to photograph all spheroids suitable for analysis (i.e. avoiding spheroids in close proximity of the walls of the wells, bubbles in the gel, or other spheroids). Subsequent evaluation was performed in a blinded manner. The information was added in the Material and Methods section of the manuscript.

Results 3.1-Line 351-352: it is not clear what histopathological parameters are being referred to here.

Response: The text was amended.

Results 3.2-Lines 412-415: orthotopic implantation of the FAP+ stromal cells did not lead to tumour formation. In this experiment were euthanised after a median follow up time of 85-140 days which is a relatively broad time range. As tumour formation was not observed, the clinical implications of tumour formation were obviously not an indication for euthanasia however it is not clear from the methods what the criteria or clinical signs that prompted the implementation of euthanasia were and whether this was considered normal.

It is also not clear from the results what was observed when examining 10 uM sections stained with HuNu. Were the stromal cells originally implanted visible but showed no signs of growth or were the cells not evident at all? How many 10 uM sections were examined per mouse and how was it determined that what was observed did not constitute tumour formation? As there are no figures or data presented for this experiment, it is difficult to determine how thoroughly analysed and how convincing this data is.

Response: For two cell cultures, tumorigenicity was evaluated after 85 and 88 days, which is the usual interval after which tumor formation can be observed with patient-derived glioblastoma cells in our model. In the remaining two experiments, the interval was (somewhat arbitrarily) extended to 109 and 140 days, respectively, to reveal possible delayed tumor formation. The mice exhibited no clinical signs of intracranial tumors at the time of sacrifice.

To detect the presence of the tumors, several sections were stained using Differential Quik Stain Kit as the brains were being cut. This method reliably identifies the presence of a tumor mass in xenografted tumors in our models. Since no tumors were revealed, immunohistochemistry to visualize human nuclei was used. At least three sections around the implantation site (which could be readily identified by scaring of the tissue) were stained for each mouse. With the exception of one cell culture, where individual human cells remained in the immediate vicinity of the implantation site (but did not form a tumor mass or spread into the surrounding tissue during 109 days), no human cells were found in the brains.

The corresponding sections in Material and Methods and Results were amended to provide more details about the tumorigenicity experiments.

Results 3.4-It is not clear that the assumptions for performing a Student T-test (Fig 4A, 5A) or ANOVA (Fig 4B, 5C) with regards to normality and homogeneity of variance have been met. Statistical analyses should confirm that the appropriate tests eg. Shapiro-Wilks, Levene’s etc) have been employed to confirm the assumptions are complied with.

Response: The data have no significant departure from normality as confirmed by Shapiro-Wilk test and normal p-plots for values from individual visual fields and means for individual inserts (five visual fields were evaluated per one insert). Student t-test was incorrectly mentioned in the original text when in fact a t-test with separate variance estimates (Welch’s t-test) was used as it does not assume that the two populations have the same variance. This was corrected in Figure 3A and 4A in the revised manuscript.

When reanalysing the data, Levene’s test revealed inequality of variances for data presented in Figure 4B. We therefore reanalyzed the data using a Kruskall Wallis test, which is however known to have lower statistical power. This analysis revealed that only the effect of HBVP is statistically different from control. We modified the text in the Results accordingly, the overall meaning has not been changed.

Original: „While HBVP statistically significantly increased the growth of endothelial cells, the effect of FAP+ mesenchymal cell cultures from GBMs was variable. Two of the cultures statistically significantly decreased endothelial cell growth, two had no effect, and one had a statistically significant pro-proliferative effect (Figure 4B).“

New: „While HBVP statistically significantly increased the growth of endothelial cells, the effect of FAP+ mesenchymal cell cultures from GBMs was variable and not statistically significantly different from control (Figure 4B).“

Section „2.13 Statistical analyses“ in Materials and Methods was amended.

Results 3.6/Figure 8: text and figures are inconsistently labelled as either 27A or 27B. Please correct.

Response: Corrected, thank you for noticing this inconsistency.

Results 3.7-Line 601-602: it would be more accurate to state ‘we identified two parameters which correlate with progression’ than ‘we identified two parameters which influence progression’.

Response: Corrected.

Eva Balaziova, Aleksi Sedo, Petr Busek on behalf of the authors

Reviewer 3 Report

This is a very interesting manuscript focused on interactions between stromal cells in GBM and the impact of a specific FAP+ subpopulations and presenting original data. The authors have previously demonstrated that FAP (Fibroblast activation protein), a protein marker of cancer-associated fibroblasts (CAFs) and tumor-associated mesenchymal cells is increased in GBMs In the current study they addressed the role of FAP+ stromal cells in the GBM microenvironment and GBM progression. They performed a comprehensive survey of FAP expression in GBMs (using double immunofluorescence labeling) and glioma cell lines, and analyzed FAP influence on various interactions of glioma cells with other cells. They found a quantity of FAP+ stromal cells positively correlated with tumor vascularisation and microvascular proliferations but there was no significant association between FAP+ stromal cells and proliferation activity (Ki-67 labelling index) or the presence of necroses in GBM. They further explored potential link with vascularization testing events in co-cultures in vitro. used some biochemical assays to study those interactions, developed FAP+ cell cultures from GBMs to analyze their impacts or the effects of conditioned media on glioma growth. They also derived primary microvascular endothelial cell cultures from fresh human GBM tissue, which is an interesting approach making all experimental set up more relevant. They tested FAP+ mesenchymal cells increase the growth and migration of glioma cellsusing two glioma cell lines (U251, U87) and conditioned media from 497 FAP+ mesenchymal cell cultures (31A, 33A, 34A, 36A, 38B) and from human brain vascular pericytes (HBVP), respectively.

In general, the assays and methods are appropriate May be chicken chorioallantoic membrane assay is historical and out fashion, nobody use it anymore, but they complemented study with 3D angiogenic sprouting assay to support their claims, so it is appropriate. A number of samples/replicates is sufficient for this type of studies. Method description is adequate and sufficient.

The manuscript is well written and illustrated.

Minor comments:

  • The information about statistical analyses is very general. I did not find information about a number of biological replicates or numbers of independent experiments for biochemical part (i.e. in Fig.4 they wrote about a number insert, but not about a number replicates, independent experiments). I do not doubt the conclusion because they tested several FAP+ cell cultures but it is a standard to provide with such information. In fig.6 it is hard to extract information about reproducibility of this experiment.
  • They present the results from two independent experiments performed in quadruplicates. In fact it would better to present data form 3 independent experiments, each in triplicates, as this is a standard.

Author Response

We appreciate the reviewer’s constructive suggestions. We have made the corresponding changes in the revised manuscript.

Point by point response:

In general, the assays and methods are appropriate May be chicken chorioallantoic membrane assay is historical and out fashion, nobody use it anymore, but they complemented study with 3D angiogenic sprouting assay to support their claims, so it is appropriate. A number of samples/replicates is sufficient for this type of studies. Method description is adequate and sufficient.

The manuscript is well written and illustrated.

Minor comments:

  • The information about statistical analyses is very general. I did not find information about a number of biological replicates or numbers of independent experiments for biochemical part (i.e. in Fig.4 they wrote about a number insert, but not about a number replicates, independent experiments). I do not doubt the conclusion because they tested several FAP+ cell cultures but it is a standard to provide with such information. In fig.6 it is hard to extract information about reproducibility of this experiment.
  • They present the results from two independent experiments performed in quadruplicates. In fact it would better to present data form 3 independent experiments, each in triplicates, as this is a standard.

Response: We regret the lack of detail information about the statistical analyses in the original manuscript, we have included more detail in the revised version. The limited amount of patient-derived FAP+ mesenchymal cells and primary endothelial cells from glioblastomas (only cell cultures at low passages were used to prevent phenotype shift, cells grew rather slowly to relatively low densities, part of the cells was used for characterization and other functional studies) was an obstacle for performing multiple independent experiments with each patient-derived culture. We have tried to mitigate this limitation by performing the assays for several independent cell cultures isolated from different glioblastoma patients and when more material was available, we preferred to use the particular cell culture for additional analyses rather than repeating the experiment. When experiments were repeated for some of the cultures, the results were reproducible.

The number of inserts in Figure 3 was corrected to “replicates”. The figure summarizes data for individual cell cultures evaluated in independent experiments. In spite of the limited availability of the cells, we were able to perform the experiment twice for cell culture 19B with highly similar results. For the remaining cultures one experiment was performed in at least triplicates, for each of the replicates five independent visual fields were evaluated.

Fig.6 A summarizes results for conditioned media from individual cell cultures evaluated in several independent experiments. Again, due to the limited availability of the cells, medium from each FAP+ mesenchymal culture was tested in one experiment. Two independent wells, each containing 10-20 spheroids were used per treatment. For HBVP conditioned media, two experiments were performed with comparable results, a representative experiment is shown in the manuscript. Some of the media from FAP+ mesenchymal cells (27B, 40A, 42A) were further tested in independent experiments using mixed spheroids containing HUVEC and HBVP (these data are not shown in the manuscript as they did not bring additional information compared to spheroids containing only HUVEC) with a similar increase in the number of sprouts compared to control conditions. These results support the reproducibility of the proangiogenic effect of the conditioned media in the 3D assay.

The fact that the reported effects were consistently observed for different cell cultures from different glioblastoma patients supports in our opinion the conclusions of the study.

Information on the number of replicates and independent experiments was amended and a short paragraph was added at the end of the discussion summarizing the possible limitation of the study.

Eva Balaziova, Aleksi Sedo, Petr Busek on behalf of the authors